# Turning the Tide: An Analysis of Factors Influencing the Adoption of Biodiversity-Enhancing Measures on Agricultural Land at the German Baltic Coast

Kathleen Schwerdtner Máñez [1,2,*], Wanda Born [1,3] and Susanne Stoll-Kleemann [1]

1 Institute of Geography and Geology, Department of Sustainability Science and Applied Geography, University of Greifswald, 17489 Greifswald, Germany; wanda.born@uni-greifswald.de (W.B.); stollkle@uni-greifswald.de (S.S.-K.)
2 International Institute Zittau, TUD Dresden University of Technology, 02763 Zittau, Germany
3 DAUCUM—Werkstatt für Biodiversität, 14471 Potsdam, Germany
* Correspondence: ksmanez@gmail.com

**Abstract:** The agricultural sector plays a major role in turning the tide of biodiversity loss. In the European Union, land use decisions and biodiversity are strongly influenced by the Common Agricultural Policy (CAP). Despite massive investment in subsidies to incentivize environmentally friendly farming practices, the CAP has so far failed to preserve the biodiversity of Europe's farmland. A simplistic understanding of farmers' motivations, dominated by rational, economic cost–benefit considerations, is one of the reasons for this failure. Our study contributes to this discussion through the identification of factors influencing the decision making of farmers. Through a case study approach in a biodiversity-rich region on the German Baltic coast—the so-called hotspot 29—we classify a number of personal, social/group, and external factors relevant to the implementation of biodiversity-enhancing measures on agricultural land. Applying a model of pro-environmental behavior, we illustrate the interlinkages between factors and outline potential solutions to support biodiversity-enhancing behavior.

**Keywords:** agriculture; behavior change; conservation; policy interventions; pro-environmental behavior

## 1. Introduction

Biodiversity is declining faster than at any time in human history, with up to one million plant and animal species on the brink of extinction [1]. Over the past 50 years, the conversion of natural ecosystems into farmland and decades of the "cheaper food paradigm", which aims to produce more food at lower cost, has made our global food system the main driver of the biodiversity crisis. Agriculture is also a major contributor to global greenhouse gas emissions and adds to climate change, thereby further degrading habitats and causing the dispersion of species to new locations [2]. Half of the world's land is now used for agriculture, and in the European Union, about 40% of the land is cultivated [3]. As a result, the agricultural sector has a major role in turning the tide of biodiversity loss [4].

In Europe, a long tradition of arable and livestock farming has created a rich biodiversity specifically adapted to agricultural landscapes [5]. However, policies and economic structures in support of the "cheaper food paradigm" caused considerable change in agricultural practices, with major impacts on biodiversity [6]. Current production systems depend heavily on the use of fertilizer, pesticides, energy, and water, and on unsustainable practices such as monocultures and heavy tillage. This has reduced the diversity of landscapes and habitats, threatened or destroyed breeding, feeding, and/or nesting sites for birds, mammals, insects, and microbial organisms, and displaced many native plant species [2]. Biodiversity-relevant structures such as small water bodies, vegetated

patches, or hedges were removed, causing habitat loss and preventing wildlife from moving between the remaining natural areas. Many landscapes were converted into intensively farmed uniform large-scale feedlots, which are a major threat to farmland wildlife such as birds and insects [7–9]. For example, traditionally managed semi-natural grasslands, which support the exceptional local richness of many taxa such as plants, fungi, and insects, have declined by more than 95% over the last century [10].

A number of publications, including high-level policy reports, have highlighted the importance of transforming food systems as levers for biodiversity conservation [2,4,11]. Such a transformation needs to address the entire food supply chain from production to consumption, which requires manifold societal and political changes, for example, a shift toward more plant-based dietary patterns [2] or agricultural policies providing subsidies with a strong link to conservation outputs [5]. As most direct pressures on biodiversity exist at the production level, there are good reasons for moving sustainable agriculture to the top of the conservation agenda [11].

Land use decisions and biodiversity are strongly influenced by the European Union's Common Agricultural Policy (CAP). Various instruments to reverse biodiversity loss and reduce the negative environmental impacts of agriculture exist, with mandatory cross-compliance criteria for farmers who receive support and additional voluntary measures such as agri-environmental programs [12]. But despite a massive investment in subsidies to incentivize environmentally friendly farming practices, the CAP has failed to preserve the biodiversity of Europe's farmland [12,13].

One of the reasons for this failure is a simplistic understanding of farmers' motivations, dominated by rational, economic cost–benefit considerations [14,15]. There is increasing evidence for a far more nuanced and diverse decision-making behavior [16,17], and recent research on the role of agriculture in biodiversity conservation has applied a number of methodological approaches, for example, the theory of planned behavior [18] for analyzing the conservation of on-farm biodiversity [19], the role of intrinsic motivations and barriers for adopting conservation practices [20], or the relation of farmer's mental models to their land management practices [21,22].

Our article contributes to this growing discussion through a case study in which we identify factors influencing the implementation of biodiversity-enhancing measures on agricultural land. The case study took place in a biodiversity-rich region on the German Baltic coast—the so-called hotspot 29. Hotspot 29 is a unique, multi-faceted coastal landscape in the state of Mecklenburg-Vorpommern, Germany, with bodden, creeks, islands and peninsulas, flat and steep coasts, salt marshes and reeds, bogs and alder forests [23]. The area extends over 1210 km$^2$, of which 2/3 are under nature conservation. A total of 42% of the entire hotspot is agricultural land, of which ¼ is protected as FFH areas under the EU Habitats Directive. Mecklenburg-Vorpommern is characterized both by a high proportion of arable land and the size of its farms. With 274 hectares, the state average is significantly above the national average of 63 hectares. A total of 28.6% of the land is managed by farms with low-input farming, 43.2% by farms with medium-input farming, and 28.2% by farms with high-input farming [24]. Within hotspot 29, the average size of farms is approximately 490 ha, and thereby well above the average for Mecklenburg-Vorpommern. Every 8th farm cultivates between 1000 and 3800 ha. The huge cultivated areas and lack of connecting structures cause habitat fragmentation and hinder the movement of species. Sustainable use, appropriate restoration, and biotope connecting measures could improve the conservation status of this coastal landscape and preserve the region's outstanding biodiversity in the long term. Recent research proves the importance of diversifying cropland and reducing field size in order to create a landscape-level mosaic of natural habitat patches and fine-grained cropland diversification [25]. However, not enough farmers are implementing biodiversity-enhancing measures on their land. As a result, valuable habitats and many of the hotspot's native species are under threat.

Our hypothesis was that the current agri-environmental program in Mecklenburg-Vorpommern does not sufficiently take factors beyond cost–benefit considerations into

account. We predicted that through a case study approach, we will identify factors currently acting as barriers to the implementation of biodiversity-enhancing measures. This will allow us to suggest options to address them, including improved measures.

During our research between 2021 and 2022, the CAP was under transition from the old funding period (2018–2022) to the new funding period (2023–2027). Measures under the new CAP period had just started. As part of eco schemes under the first pillar, Mecklenburg-Vorpommern offers the support of non-productive areas, flowering areas and strips, strips or areas of old grassland, crop diversification in arable farming, agroforestry, extensification of individual business branches, result-oriented extensive cultivation, and no synthetic chemical plant protection. Under the second pillar, the following measures are supported: conversion of arable land into permanent grassland, moor conservation damming and paludiculture, riparian strips, environmentally friendly fruit and vegetable growing, erosion control areas, extensive permanent grassland management, salt grasslands and shorebird breeding areas, buffer strips at legally protected biotopes, and avenues and forest edges, among others (see https://www.landwirtschaft-mv.de/static/LFA/Inhalte/Fachinfo/FP_MV.pdf, accessed on 21 November 2023). Under the last funding period, the state paid EUR 276 million to support organic farming, and another EUR 225.7 million for agri-environmental and climate protection measures [26]. An evaluation of the impacts of the funded measures on biodiversity has not been made, and the effects of such measures in general on slowing the biodiversity decline are largely unknown [27]. While our work does not offer an impact evaluation either, it may support the acceptance and uptake of biodiversity-enhancing measures by developing potential solutions that could be integrated into the measures of the upcoming funding period. In November 2023, the Associations Platform on the Common Agricultural Policy (CAP), a group of almost 40 organizations from agriculture, environmental protection, nature conservation, climate protection, animal welfare, consumer protection, and development cooperation, presented a paper with goals, demands, and necessary steps for the future EU agricultural policy. In it, the organizations call for a fundamental change in strategy and policy in European agricultural policy and its implementation in Germany. They also argue that the current funding period until 2027 must be used as a transition phase for the long-term transformation of the CAP [28].

This article is structured as follows: first, an interdisciplinary literature review provides the background on factors that influence land use-related behavior. Then, the methodological section introduces our research and analysis framework and is followed by the results section. In the final section, we discuss our results and suggest potential solutions.

## 2. Literature Review on Factors Affecting Farmers' Decision-Making Behaviors

### 2.1. Personal Factors

Personal factors are individual characteristics that influence the behavior of a person, such as personality traits, values, attitudes, and objectives [29]. Together with environmental awareness, they play an outstanding role as motivating factors in the adoption of conservation measures [30]. For example, recent work has proven the significance of openness and conscientiousness in farmers' decisions [31].

Environmental awareness and moral concerns are linked to a person's values and attitudes. Values can be understood as "desirable transsituational goals, varying in importance, that serve as guiding principles in the life of a person or other social entity". Attitudes can be defined as a disposition to react favorably or unfavorably to an object, person, institution, or policy. They are evaluative, that is, directed at a given object or target, for example, that landscapes are to be cultivated and shaped. Although attitudes are assumed to be relatively stable, evaluations can change rapidly as events unfold and new information becomes available [18].

Objectives are the goals that farmers pursue through their activities [32]. Farmers may have conflicting objectives, such as creating wealth and embracing conservation [33]. Cognitive dissonance describes a situation in which mismatches among cognitive elements require a person to choose between two incompatible beliefs or actions [34]. A farmer

might care about the environment, but at the same time, feel that pursuing this goal will undermine economic stability.

Cognitive factors describe the way a person learns and performs, and understands and perceives things. Farmers may perceive a certain measure as difficult and risky, whereas another measure is seen as easy and non-risky [32]. This risk perception is significantly related to both the extent to which farmers adopt conservation practices and to the types of conservation practices they choose to adopt [33]. Perceived costs and benefits—which are different from objective measurements—also play an important role [35]. For example, Davidson et al. [36] have shown that expected improvements in soil quality and biodiversity are important motivations for the adoption of certain agricultural practices. Knowledge is another cognitive factor that includes both actual skills as well as information about certain opportunities, such as available support. A study in five EU countries identified a lack of knowledge as the main reason for not participating in agri-environmental schemes [37]. And finally, perceived behavioral control or self-efficacy also belongs to this category. It is defined as the "perceived ease or difficulty of performing the behavior" [38]. People who feel they have the self-efficacy to carry out a certain behavior are more likely to do so than people who perceive themselves as lacking the ability to behave in the desired way [39].

Emotions are increasingly seen as a major factor for all environmental-related behaviors and to understand the relationship between people and their environment [40]. Being aware of the consequences of one's unsustainable behavior can cause emotionally distressing reactions, such as shame or guilt. In addition, studies show a strong connection between feelings of guilt and responsibility to the willingness to make sacrifices for the environment [41]. On the other hand, positive emotions such as pride, hope, and gratitude are known to be key drivers for action [42,43]. Several authors have called for the systematic inclusion of emotional elements in environmental behavior choices [44–46].

### 2.2. Social and Group Factors

Humans are social beings. Our identity, our way of being in the world, is deeply influenced by the people around us [34]. Social factors refer to this interaction including variables such as social norms, which prescribe or proscribe certain behavioral options [47]. Norms can be a powerful force in both inhibiting and encouraging pro-environmental behavior and pro-environmental behavioral change can be considered as a transition in social norms [34]. They are important for group coherence and mediate between the identity of the individual and that of the group [48]. Groups give a sense of social identity and a sense of belonging to the social world. The central hypothesis of the social identity theory is that members of an in-group ("us") will seek to find negative aspects of an outgroup ("them"), thus enhancing their self-image [49]. According to social representation theory, social groups have culturally and historically contingent systems of values, ideas, and practices [50]. If confronted with novel, unknown, or potentially threatening phenomena, existing ideas and practices are used as collective coping mechanisms [51]. In their review, Ahnström et al. [52] argue that there are many social identities among farmers, such as milk producers, cereal growers, or organic farmers, either defined by farmers themselves or the surrounding society. A careful analysis of the respective social group is required to understand and potentially influence behavior. While group behavior is driven by descriptive norms, which ensure conformity with the behavior of other group members, injunctive norms reflect the expectations of society. Improving their public image has been shown to be an important factor behind the adoption of more sustainable practices, such as organic farming [35]. The individual sensitivity of a farmer to conformism and moral pressure determines how strongly both types of social norms will impact conservation-related decisions [53].

### 2.3. External Factors

The Common Agricultural Policy (CAP) is the main policy framework for the European Union and its member states in the design and implementation of both mandatory

and voluntary agri-environmental policy instruments. Offering direct income support in the form of subsidies, CAP-based policies and regulations have significant steering effects on farming decisions. The first pillar, market management and income support, provides hectare-based payments that require compliance with (basic) environmental requirements [54]. The second pillar, focusing on rural development, forms the basis for agri-environmental schemes, which is the most important instrument for biodiversity conservation on agricultural land. Other political factors such as laws and regulations determine the limits under which a farmer can operate. For example, farmers are required to adapt fertilization to nitrogen levels in the soil and to confirm these levels regularly.

Economic aspects strongly influence decisions. Farmers may perceive it to be easier to earn money by producing high yields with few cash crops, and harder to increase profitability by enhancing biodiversity through a broader mix of crop species. To make the farm business profitable, therefore, puts agricultural production before conservation [52]. Other factors such as the availability of markets or consumers for certain products also play a role.

Socio-demographic factors have been shown to significantly influence the uptake of conservation measures on agricultural land [55]. For example, a review of 279 studies on the adoption of agricultural conservation practices in the United States identified formal education as one of the relevant variables and an important predictor of behavior [56]. In addition, a number of authors have highlighted the positive relationship between the level of agricultural training and the adoption of conservation agriculture [57,58]. While other factors, such as age and gender, also play a role, their influence seems to be less significant. For example, research by Serebrennikov et al. [59] shows that farmers' age systematically influences organic farming adoption, but not the adoption of other measures. A meta-analysis of 125 articles by Mozzato et al. [60] finds that younger farmers prevail as early adopters, while older farmers act as followers. The same authors also argue that female farmers have a higher motivation to adopt environmentally friendly farming practices, especially in Europe and North America. In his review, Burton [61] concludes that demographic characteristics are influenced to varying extents by cultural–historical patterns leading to cohort effects or socialized differences. This is also confirmed by Mozzato et al. [60], who highlight the influence of the geographical context as well as temporal trends on both socio-demographic as well as other factors.

Farm-specific factors such as farm size and local environmental conditions, land tenure, and the type of farming that is practiced have been shown as influential. According to Rodriguez Entrena and Arriaza [58], farm size is positively correlated to the adoption of conservation agriculture, while Traoré [62] did not find such a correlation. Ownership of land has been shown to have an influence on the uptake of conservation measures ([60] and references therein). However, a more detailed analysis by Leonhardt et al. [63] reveals that instead of a simple distinction between rented and owned land, specific aspects such as tenure security, lengths of the contract, or the strength of the relationship between the landowner and the tenant play a role.

## 3. Methodological Approach

Our methodology consists of three components. As a conceptual frame, we applied a case study methodology. Case studies are one of the most extensively used strategies of qualitative social research. Qualitative social research, in particular in environmental psychology, has been shown to provide major insights into how respondents relate to commonly used theoretical constructs and reveal important themes neglected in quantitative research [64,65]. Case studies aim to generate an in-depth and multi-faceted understanding of a complex issue in its real-life context [66]. Studied cases are bound by time and activity, and a variety of data collection procedures such as questionnaires or in-depth interviews are typically used over a sustained period of time [67]. Case studies looking for causal factors to explain decision making are classified as explanatory [66]. The aim of our explanatory case study approach was to answer the following questions:

1.  Which factors play a role in implementing biodiversity-enhancing measures on agricultural land?
2.  What are possible solutions to remove the identified implementation barriers?

Our second component was semi-structured interviews with farmers, for which we developed guidelines (see Supplementary Material). Contact details of farmers in hotspot 29 were researched and potential interviewees were contacted by phone and/or mail. Of 40 contacted farmers, 13 agreed to be interviewed. Some of the interviews were recorded digitally (n = 5), but the majority were documented via transcripts only at the request of the interviewees.

Our third component was the application of an already successfully adapted behavior change model (see Figure 1), originally based on the model of pro-environmental behavior developed by Kollmuss and Agyeman [68]. Examples of previous research can be found in Stoll-Kleemann and Schmidt [39] and Stoll-Kleemann [44]. The reasons for choosing this conceptual model lie in its comprehensiveness and its multifactor approach. Furthermore, according to Gifford and Nilsson [69], many studies have shown that well-known and established social psychological models that would, in general, be appropriate here, such as the theory of planned behavior [38], the value-belief-norm model [70], and the norm activation theory [71], should be "expanded to include other personal and social factors" (p. 141). The underlying structure and the factors of the model are based on the literature review but have been redefined with the results of the semi-structured interviews. A qualitative content analysis was used to generate codes from the answers. These codes were matched with the previously identified factors, and, if necessary, the factors were rephrased.

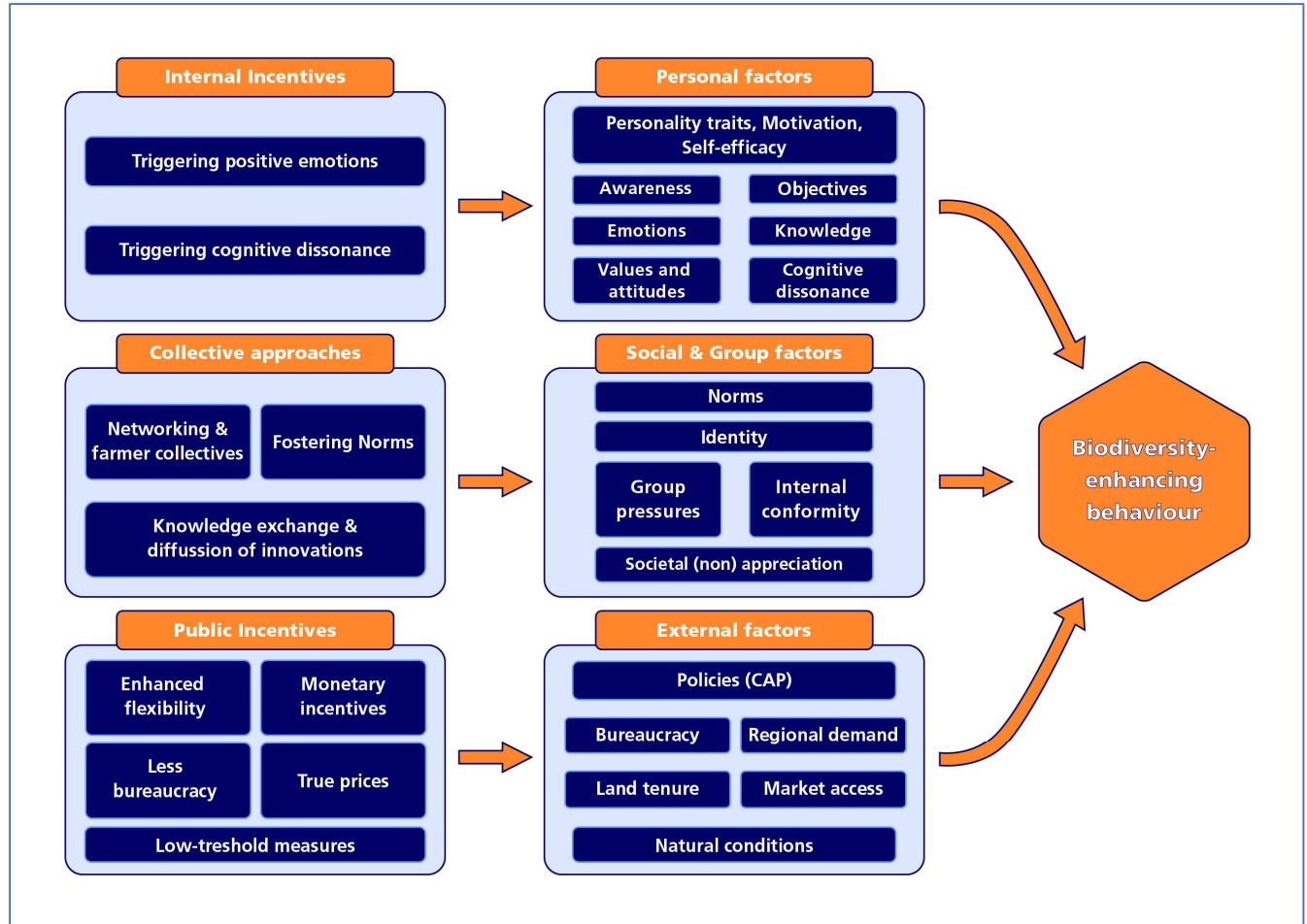

**Figure 1.** Model of biodiversity-enhancing behavior. Adapted from Stoll-Kleemann [44].

Stoll-Kleemann's original model was applied to a very specific target group, namely farmers, for an equally very specific environmentally friendly behavior, which is biodiversity-conserving or even enhancing behaviors in the context of professional behavior in the agricultural sector. In particular, the target group was narrower than in the previous model application areas, which focused on consumers and citizens in relation to, for example, a plant-based diet or ocean-friendly behaviors. It is, therefore, not surprising that there are no fundamental deviations, but there are special characteristics that are reflected in the "external factors" section. For example, CAP plays a special role, including land tenure and natural conditions, as it is logical for the agricultural sector, but the excessive and often counterproductive bureaucracy also proves to be an additional barrier to biodiversity-enhancing behavior.

The model is divided into personal factors, social and group factors, and external factors (such as a political frame with a focus on CAP, economic factors such as market access, or issues such as administrative implementation). In order to select personal factors, we examined the influences of values/attitudes, problem awareness, and biodiversity knowledge on potential behavior. Also important are more general personal factors such as personality traits, motivation, self-efficacy, and habits. We also looked carefully at the role of emotions (such as frustration and resignation) and cognitive dissonance, which are core factors that potentially help us to understand the personal reluctance to avoid the biodiversity-enhancing behavior of farmers. The interrelations between emotions/cognitive dissonance and knowledge, values, and attitudes are also explicitly incorporated into the model, which increases its complexity but also its explanatory power. In addition, we address social and group factors such as social identity, social norms, group pressure, and internal conformity in guiding biodiversity-enhancing behavior. These factors can either support biodiversity-enhancing behavior or act as barriers, which is why they are positioned in the middle of the model. In order to positively influence these factors and overcome the barriers, different internal (such as targeting emotions) and external incentives (such as true prices) as well as collective responses (e.g., increased networking among farmers)—explained in detail in the discussion section—are suggested.

## 4. Results

The 13 interviewed farmers cultivate 11,500 ha in total. Eleven of the respondents are full-time farm managers, while two run a farm but produce a niche product. The farms are primarily managed by men (11 out of 13), and most of the farm managers have studied (n = 7). Almost half of the interviewees (n = 6) combine organic or biological cultivation methods with conventional agriculture. Six farms are managed exclusively using conventional agriculture methods, and two are exclusively managed according to EU organic standards, including special forms (horse husbandry and beekeeping). The surveyed cash crop farms primarily produce grains such as winter wheat, winter barley, or maize. An overview of the investigated farms is given in Table 1.

**Table 1.** Farm characteristics of investigated farms.

| | Occupation | | Operating Structure * | | | | Ownership of Land | | |
|---|---|---|---|---|---|---|---|---|---|
| | Main Occupation | Sideline Business | Crop Specialists [1] | Livestock Specialists [2] | Mixed Farms | Non-Classifiable | Freehold | Leasehold | Both |
| Number | 11 | 2 | 2 | 1 | 7 | 3 | 0 | 1 | 12 |
| % | 85% | 15% | 15% | 8% | 54% | 23% | 0% | 8% | 92% |

* According to agri-environmental indicator "farm specialization" (2020), source: https://ec.europa.eu/eurostat/statistics-explained/index.php?title=Farms_and_farmland_in_the_European_Union_-_statistics, (accessed on 17 November 2023). [1] Including general field cropping and cereals, oilseed, and protein crops. [2] Including pork production.

The land farmed by our respondents accounts for approximately 23% of the hotspot area and almost 50% of its agricultural land. Although the number of interviewed farmers is rather small, their decisions affect a considerable part of the entire hotspot. The following

section introduces the results from our interviews and answers the question of which factors play a role in implementing biodiversity-enhancing measures on agricultural land. An overview of the relevance of each factor is presented in Table 2.

**Table 2.** Relevance of factors among interviewees.

| Relevant Factors | Number of Interviews in Which the Factor Was Relevant | Percentage |
|---|---|---|
| Personality traits | 10 | 77% |
| Motivation | 11 | 85% |
| Self-efficacy | 11 | 85% |
| Awareness | 12 | 92% |
| Values and attitudes | 12 | 92% |
| Objectives | 10 | 77% |
| Cognitive dissonance | 5 | 38% |
| Emotions | 8 | 62% |
| Knowledge | 10 | 77% |
| Social norms | 5 | 38% |
| Identity | 5 | 38% |
| Group pressure | 3 | 23% |
| Internal conformity | 3 | 23% |
| Societal (non-)appreciation | 5 | 38% |
| Natural conditions | 10 | 77% |
| Policies | 12 | 92% |
| Bureaucracy | 11 | 85% |
| Land tenure | 8 | 62% |
| Regional demand | 4 | 31% |
| Market access | 6 | 46% |

*4.1. Personal Factors*

We found evidence for the <u>motivation</u> to conserve biodiversity among almost all respondents. Statements such as "*The farmer here* (in this landscape) *becomes a conservationist, service provider, and landscape manager*" clearly show this. One farmer said, "*I am already thinking ahead—at least I think so. Due to climate change, it is getting warmer and warmer, so we grow crops that can still flower in autumn, i.e., again for insects and biodiversity*". In this quote, the link between <u>awareness</u> of the impact of climate change on biodiversity and openness (personality traits) and <u>motivation</u> toward alternative ways of farming is clearly made.

While the respondents had favorable <u>attitudes</u> toward biodiversity conservation, they also struggled with contradictory objectives. Statements such as "*We try to find a compromise between economy and ecology*" and "*(biodiversity-enhancing) measures have to pay off*" illustrate the perceived contrasts between biodiversity protection with low revenues on the one hand and solid profits without biodiversity protection on the other. One respondent called the payments for biodiversity-enhancing measures as part of agri-environmental schemes a "*zero number*". While farmers are very aware of their role as landscape designers, with clear evidence for <u>self-efficacy</u>, they also act as entrepreneurs. Their conflicting <u>values, attitudes,</u> and <u>objectives</u> lead them to experience <u>cognitive dissonance</u>, which is linked to strong emotional reactions.

<u>Emotions</u> play an important role. Farming is not just an economic activity, it is a way of living and a source of identity for all interviewees. Farmers feel deeply connected to their land, intending to take good care of the resources they work with. "*Our farm develops together with the land*", said one of the respondents. While our interviews also revealed pride for accomplishments, the outstanding emotion among all respondents was frustration. "*The person who wants to maintain good farming practices, his animals and land well, you just don't let him do his job properly*". This frustration is primarily caused by external factors, which will be explained in more detail in the respective section. In particular, frustration was evident when farmers talked about their experiences with local administration regarding

CAP regulations, for example, with respect to the state agency for agriculture. For example, the statement "*You talk against the wall*" describes the negative experiences of farmers.

There was a clear <u>awareness</u> among the interviewees of environmental problems such as increasing summer droughts. This awareness was largely based on personal experiences, although some interviewees also mentioned reporting in newspapers, TV, or information from farmers' organizations. These issues were generally perceived as a business challenge and addressed through an adaption of cultivation practices. For example, heat-resistant wheat varieties from France are now cultivated in the area. We found less evidence for an awareness of biodiversity loss. In particular, the decrease in insect diversity and biomass was mentioned, and the majority of farmers participated to some extent in flower strip programs. Several farmers undertook self-motivated actions for insects, such as leaving strips when mowing, or not cultivating certain places in which insect populations had developed. The success of these actions also leads to a sense of pride, as evident in the following statement: "*This field edge has developed so well that every year someone comes and counts the insects*". Several respondents explicitly linked certain ways of farming to biodiversity, as evident in the following statement: "*The less animal husbandry, the fewer insects*". Another respondent pointed out the relevance of certain cultivation forms for preserving cultural landscapes: "(Without mowing) ... *there is more and more impoverished landscapes.* (In these landscapes...) *Hardly a bird sings anymore*".

<u>Knowledge</u> about biodiversity varies considerably between the farmers. While statements such as "*Biodiversity is what you find there*" reflect a rather generic understanding of the concept, four of the respondents also explicitly mentioned habitat and genetic diversity. Knowledge of species also differs considerably, ranging from a focus on commonly known species such as "*hares, tits and deer*" to professional knowledge on migrating and wintering bird species.

While farmers are aware of their role in biodiversity conservation, all complain about (legal) restrictions hindering them from turning their <u>awareness</u> into actions. One respondent said, "*Sometimes I would like to have a different influence (on biodiversity) than regulations allow*". While biodiversity-enhancing measures are generally perceived as positive, farmers are especially careful with the implementation of measures that may develop into legally protected habitats over time. As one farmer said, "*If I were to make a decent profit, then I could also think about hedges. But I don't want to do that too much, because then I create a biotope, which then falls on my feet again*".

*4.2. Social and Group Factors*

<u>Social norms</u> define a common understanding of acceptable behavior within a group. Despite the fact that our questions did not in particular aim at identifying norms, we found evidence for the relevance of social norms. For example, interviewees argued that it is important "...*to take good care of animals and land*...".

Social norms are closely linked to <u>group identity</u>, which we understand as a function of <u>group pressure</u> and <u>internal conformity</u>. People who experience group pressure may feel the need to conform in order to fit in, even if that challenges their personal preferences. One of the respondents argued that regardless of her personal preferences, she feels pressured towards certain forms of management: "*The land ... has to look tidy. We have visitors, for example when someone from the bank comes*".

We found the perceived <u>societal (non-)appreciation</u> of agriculture in general and of the work of farmers in particular to be of major importance. The interviewees perceive low social esteem and identification with their work and complain about a lack of understanding of farm processes. One farmer said, "*It makes a real difference what you call things. If my plants are sick, I use crop protection products to support them. Nowadays, they are called pesticides and perceived as poison*".

### 4.3. External Factors

Natural conditions, such as local climate or soil conditions, have a significant impact on farm operations and farm success. A number of recent environmental changes that pose additional challenges were reported by our interviewees. In particular, increasing spring droughts resulted in yield losses in the past 3 years. Also, higher water levels in coastal salt grasslands caused delays in the start of grazing, which increased production costs for meat.

The most frequently mentioned external factor was policies, namely the Common Agricultural Policy (CAP). Although all but one farm receives basic income support, the CAP is predominantly perceived as negative. The criticism includes the complexity of regulations (*"I have put the binder on the second pillar of the CAP away for now"*), their perceived inconsistency and senselessness (*"Humus must be built up in organic farming, but horse manure is not allowed to be stored outside"*), and a general feeling of mistrust and insecurity toward changes in regulations (*"We don't really know anything anymore. There is basically a new regulation every four weeks and this has been going on for a year. The most unreliable partner in this whole story is politics"*). Missing flexibility, for example, with respect to time limits for measures, was criticized, and the fact that measures offered by the CAP do not fit with the needs of the individual farms was seen as an important barrier to more participation in voluntary measures. Farmers want more freedom for operational decisions, such as the mowing time. Also, for the measure *"Diverse crops in arable farming"*, legumes must be grown on at least 10% of the eligible arable land. For farms with several thousand hectares, this requires land reallocation of several hundred hectares for legumes only.

In addition, the risk of sanctions in the case of non-compliance was often mentioned. One interviewee reported *"We sowed a few square meters more flower strips than we had indicated in the application because there was still space. For that, part of the subsidy was deducted from the whole farm. They (the authorities) use satellite technology to check, ours can't keep up at all"*. Another farmer said *"There is always a high risk of sanctions. If you make a 1% error, you get 30% less funding"*.

From the respondents' point of view, the financial incentives are not sufficient for participation in the yield-reducing measures. Views ranged from *"Nature conservation should be done on areas with few soil points"* (on less productive land), to *"Nature conservation should become a service and be paid appropriately"*. There was a general perception that farmers are motivated to take good care of their land and have their own ideas to support biodiversity, but compulsory policies and legal regulations leave little room for innovative approaches. Additional voluntary measures would add even more pressure on operations management, without providing adequate compensation for efforts made. As one respondent put it, *"Not every handshake has to be paid for, but it has to be . . . implementable"*.

Closely linked is the factor of bureaucracy. As one respondent puts it, *"70–80% of my time is spent on office stuff, what the EU comes up with"*. While part of the problem originates in the CAP itself, local authorities involved in the implementation of land management are also seen as part of the problem. Statements like *"The communication with (local authorities) is often difficult, and you have to be very persistent"* or *"It's always about bans. They are not willing to compromise and are not solution-oriented"* appear regularly in the interviews.

Land tenure plays a significant role in the decision making of farmers. About half of our interviewees farm at least partly on rented land. The owners of the land have a considerable influence on the management of these areas. For example, one interviewee reported that his landlord had urged him to switch to organic farming. Another farmer explained that if he participated in voluntary measures and was subsidized for this work, his landlord would raise the rent. The situation is even more complicated if farmers manage the land of different owners. One of the interviewees farms 2400 ha of arable land and meadows, which are rented from 20 different landlords. While some of his contracts are long-term, others must be renewed on a short-term basis. Consequently, he has few incentives to invest in biodiversity-enhancing measures.

(Missing) regional demands and (no or limited) market access were repeatedly named as factors that hinder the possible conservation of organic or more biodiversity-friendly

ways of farming. Farmers argued that despite having an interest in producing organic milk or meat, the possibilities to market these products were extremely limited or did not exist at all. In some cases, infrastructure was also missing, for example, a slaughterhouse within a radius of less than 250 km enabling individual slaughter. It was also said that local or regional customers do not have sufficient awareness and purchasing power to buy more biodiversity-friendly but also more expensive products. While some respondents delivered their products to markets several hundred kilometers away or had contracts with large purchasers, others refrained from the production of organic products due to missing local markets. For example, one interviewee said, "*We put heart and soul into the idea and really wanted to try, but if the market is missing, it doesn't work*". Similarly, the production of organic grain is challenged because of the different conditions that apply to conventional and organic grain.

## 5. Discussion and Potential Solutions

Our results confirm an important argument from the literature: the decision-making behavior of farmers is determined by a number of personal, social, and external factors, and goes far beyond simple cost–benefit considerations [14,16]. whether or not farmers implement biodiversity-enhancing measures on their land does not only depend on their personal values and attitudes, how they feel about a particular practice, or on the knowledge they have. It also depends on the way their actions are perceived by others, or appreciated (or not) by the society at large. Authors such as Michel-Guillou and Moser [35] have highlighted the importance of improving the public image of farmers to support the adoption of more sustainable practices.

However, our case study approach also reveals a number of peculiarities that are specific to hotspot 29. One is the combination of conventional and organic farming practices on a single farm. While many studies differentiate between two different farming systems, for example, with respect to factors influencing the convention from conventional to organic farming [72,73], half of our interviewees used a mixed approach and combined conventional methods with other, more sustainable cultivation measures. This shows not only their willingness to apply such measures, but also highlights the need to identify in particular what the barriers to the application of certain measures are, and why these do not apply to others. It also supports the argument that effective policy instruments need to capture the complexity of farmers' preferences and be adaptable to context-dependent characteristics at the landscape level [74].

We also found that contrary to some common beliefs, our interviewees were highly motivated to implement biodiversity-enhancing measures, but not necessarily those offered by agri-environmental programs under the CAP. This is largely related to the perception that these measures do not fit with individual farm characteristics and management approaches. In particular, measures that require a certain percentage of the land to be managed differently are seen as critical. This is clearly linked to the large size of most farms, a specific feature of hotspot 29. Similar resistance exists for measures leading to the development of habits that would eventually fall under § 30 Federal Nature Conservation Act (BNatschG), such as the planting of hedges. Recent research indicates farmers are less likely to switch to organic production when operating on large non-fragmented plots [72].

Another peculiarity of the area was the high share of rented land, which considerably influenced the decision making of farmers. We found evidence for participation in organic farming as a result of pressure by the landlord, as well as for the inability to implement certain measures as this would lead to increasing rent or even endanger the renewal of lease agreements.

An outstanding feature was the role of external factors, including the missing regional demand and market access. Farmers willing to convert their production found it difficult to sell organic products in the region, which has the lowest purchasing power of all German states. While markets for organic products exist in cities like Berlin and Hamburg, it was

usually argued that they were too far away, or the marketing was challenged by legal restrictions, for example, with respect to the transport of meat beyond a certain distance.

What are possible solutions to remove the identified implementation barriers? Using our model, we distinguish between solutions influencing personal factors, social and group factors, and external factors.

Personal factors can be addressed via internal incentives, for example, through triggering positive emotions. There is increasing evidence that the anticipation of future emotional states—so-called anticipated emotions—plays a powerful role in shaping other-oriented behaviors [75]. While people tend to avoid taking actions that could result in negative emotions such as guilt or sadness, they pursue those that will result in positive states such as pride and joy [76]. Positive emotions assist in learning new behaviors, appreciating new and different perspectives, and expanding creativity [77,78]. Research has shown that inducing feelings of pride for positive future actions has a strong effect on pro-environmental motivation, and may be a currently under-appreciated pathway to promoting pro-environmental behaviors [76]. The US Sustainable Agriculture Research and Education (SARE) program sees the engagement of positive emotions as key to effective agricultural education toward sustainable agriculture [79]. We suggest constructing messages on biodiversity-enhancing measures that trigger positive emotions. This shall enable and empower farmers to better understand their own relationships with the natural world and to identify their own ways of contributing to the transition to sustainability according to their individual strengths, capacities, and identities. Such messages require a clear objective, must be targeted to their specific audience, describe a desired outcome, and outline an appropriate measure of success [80].

Triggering cognitive dissonance: Dissonance between the cognitions, attitudes, and behaviors that a person experiences is thought to create motivation that leads to real cognitive change [81]. Based on this assumption, cognitive dissonance interventions have been applied in an enormous number of scientific studies in social and organizational psychology [82]. Conservation organizations such as WWF emphasize the potential to induce behavioral changes through triggering cognitive dissonance [34]. Bentler et al. [82] argue that cognitive dissonance interventions represent an extremely efficient method to increase the intention for pro-environmental behavior. Bosone et al. [83] point out that people who experience cognitive dissonance also apply rationalization strategies that justify the maintenance of a behavioral inconsistency, and that a clear identification of such strategies is required to guide behavioral change interventions. While triggering cognitive dissonance might be a useful first step, it must be accompanied by other measures. Farmers might be aware of their conflicting interests and values, but feel unable to overcome them. Sufficient autonomy is a critical factor for effective cognitive dissonance interventions [82]. In particular, our results indicate that external factors such as bureaucracy or land tenure are critical and often hinder the implementation of alternative management approaches.

Addressing the identified social and group factors requires collective approaches. Research shows that fostering social norms may initiate and maintain desirable environmental behaviors [84]. Social norms are usually tied to places and the people who inhabit them, such as the immediate environment or a cultural identity. Fostering social norms, therefore, requires localized, bottom-up approaches. In order to be effective, so-called social norms interventions [85] must be based on repeated interactions with or within a group to establish new rules, copy behaviors, receive approval, and experience disapproval [84]. Targeting individuals who are most influential within a group is thought to enhance the success of such interventions [86]. If these role models adopt a new social norm that is then perceived as beneficial by others, it will spread via a small group of early adopters into the group. When sufficient positive social feedback emerges, a tipping point is reached beyond which the new social norm is seen as normal [87].

Increased networking contributes to the fostering of social norms and supports the exchange of knowledge and the diffusion of innovations. "Other farmers" are farmers' most frequently reported source of information [88], and successful farming practices

demonstrated and explained by other farmers have the strongest influence on farmers' decision making [89]. On-farm demonstrations to support peer-to-peer learning are already common practice in Europe and contribute to building social capital and networks [90]. This is key in the transition toward more sustainable agriculture, as farmers rely on collectives to find mutual support, motivation, reflection, and trust [91]. In the Netherlands, farmer cooperatives coordinating agri-environmental measures on a regional level were already established in 1991. Research shows that cooperatives play a mediating role between authorities and individual farmers and enhance farmers' motivations to participate in agri-environmental schemes [92]. A wider implementation of such cooperatives, for example, in hotspot 29, is also an opportunity for enhancing flexibility and less bureaucracy.

While having characterized farmers' collectives as part of the collective approaches, public incentives are often required to start regional initiatives. As one interviewee pointed out, "I would like to start a Landcare group here, but it's difficult for me to find enough interested people". Such joint approaches to the preservation of agricultural landscapes also support more flexibility and less bureaucracy and can be funded under the European Agricultural Fund for Rural Development (EAFRD). Each member state makes its own arrangements to support collaborative approaches. In Germany, several so-called Landcare groups exist that are involved in landscape management and habitat and species management. Their success depends on a number of prerequisites, such as shared problem awareness, good communication, access to high-quality advice and support, and third-party facilitation [93]. These are clearly linked to the before-mentioned aspects of increased networking and knowledge exchange, but also highlight the need for external support.

Monetary incentives are important for the uptake of biodiversity-enhancing measures. Insufficient funding has been named as one of the major reasons for the CAP's failure on biodiversity [94]. Our findings are in line with those of other authors who argue that the transition to biodiversity-friendly farming requires additional financial incentives [95]. However, in addition to the amount of funding, the method of payment is also important. Most payments are action-based; they are awarded for actions that are expected to achieve specific environmental outcomes. However, Member States may also provide outcome-based subsidies, which are linked to the achievement of environmental outcomes [54]. Results-based payment schemes provide attractive incentives for remunerating ecosystem services [96].

Recent research shows that an internalization of external costs into the market price of agricultural products (true costs) closes the gap between organic and conventional products. Including the positive or negative impacts of a produced commodity into their production costs at least partly corrects current market distortions, especially for cereals, most oilseeds, and most animal-based commodities [97]. This would help in addressing factors such as the missing regional demand, which results from price distortions in favor of conventional products. Michalke et al. [98] implemented an informational campaign in a German supermarket displaying products with two price tags: one of the current market price and the other displaying the "true" price, which includes several environmental externalities calculated. While their results show a persistent knowledge gap with respect to the ecological damages caused by food production, they also found that the overwhelming majority of consumers would reduce animal products and increase organic products in their food consumption if prices rose through the implementation of true costs. The successful implementation of true prices requires a societal transformation involving actors across the entire agri-food system. Policy instruments such as product labeling and differential taxes and subsidies to incentivize healthy diets and discourage food waste are needed as part of such internalization measures, together with education, information, and nudging [99].

Low-threshold measures represent an entry opportunity into nature conservation. These include, for example, deadwood and read stone structures, which are by-products of normal farming and at the same time represent valuable habitats for reptiles and insects. Rarely mown lawns on the farmstead or long-accepted break-off edges in transition areas from fringe structures to fields can also promote biodiversity, especially of insects. Charac-

teristic of such low-threshold measures is the low investment effort—all that is really needed is a little goodwill and the willingness to leave (smaller) parts of the landscape "untidy".

In addition to these low-cost and ecologically valuable measures, eco-regulations with a more exploratory character could initially be implemented in smaller areas than before. For example, the measure "Diverse crops in arable farming" in Mecklenburg-Vorpommern is only promoted if legumes are cultivated on at least 10% of the arable land. This is a barrier to entry, especially for large farms. However, the farmers surveyed showed a clear interest in cultivating legumes in smaller areas, enabling them to test the measure first before making a decision.

Addressing the biodiversity crisis will require a significant expansion of biodiversity-enhancing measures on agricultural land. An important aspect of achieving this is a better understanding of the factors that influence farmers' decision making. Our article confirms the complex nature of decisions to adopt measures, in which economic considerations are only one of many factors. It also demonstrates the usefulness of a case study approach, which allows for a nuanced and in-depth understanding of regional and other specificities. We need solutions that are not only beneficial for biodiversity, but also socially acceptable and workable. This requires better communication with farmers and, in some cases, co-development of measures. Biodiversity conservation on farmland will only be successful if it is treated as a shared responsibility between society and farmers. As the first considerations and meetings concerning the next CAP funding period are currently taking place, we hope that our results will be taken up into the design for new biodiversity-enhancing measures.

**Supplementary Materials:** The following supporting information can be downloaded at: https://www.mdpi.com/article/10.3390/su16010317/s1.

**Author Contributions:** Data collection, K.S.M., W.B. and S.S.-K.; methodology, K.S.M., W.B. and S.S.-K.; data analysis, W.B. and K.S.M.; and writing K.S.M., W.B. and S.S.-K. All authors have read and agreed to the published version of the manuscript.

**Funding:** This research was funded within the Federal Programme for Biological Diversity by the Federal Agency for Nature Conservation with resources from the Federal Ministry for the Environment, Nature Conservation, Nuclear Safety and Consumer Protection (FKZ 3520685G34).

**Institutional Review Board Statement:** Ethical review and approval were waived for this study due to the fact that each respondent was asked individually for permission to carry out an interview. Furthermore, the qualitative nature of the study did not allow for pre-defined questions which could have been approved.

**Informed Consent Statement:** Informed consent was obtained from all subjects involved in the study.

**Data Availability Statement:** The data presented in this study are available upon request from the corresponding author. The data are not publicly available due to privacy reasons.

**Conflicts of Interest:** The authors declare no conflicts of interest.

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
