# Peer review of "Turning the Tide: An Analysis of Factors Influencing the Adoption of Biodiversity-Enhancing Measures on Agricultural Land at the German Baltic Coast"

_sustainability, doi:10.3390/su16010317_

Round 1
Reviewer 1 Report
Comments and Suggestions for Authors
Dear Author,
The paper represents a very significant topic related to the acceptance of biodiversity measures.
However, the number of interviewees is small. The paper lacks numbers in terms of the concrete percentage (or number) of interviewees who gave this or that answer. It would be clearer if the results were presented in a table or graph, keeping in mind that the number of interviewees is very small. Lack of direct measures that could be acceptable or not for farmers related to the biodiversity and the environment. The discussion needs to be improved to present or emphasize what was the biggest problem for farmers - for example, costs - what costs? Seeds, fuel? Or time? - for soil preparation, for sowing? Biodiversity with only legumes included is a very poor way to improve that. However, even with that, authors should give more details - what legumes will be involved, etc.

Author Response
We appreciate the careful review by this reviewer and feel that the comments have been very helpful in order to improve the manuscript.
We are aware of the small number of respondents – however, we would like to point out that we applied a qualitative approach without aiming to find statistically significant results. We approached 40 farms which is 50% of all farmers within the hotspot area, and only 13 farmers agreed to be interviewed. However, as result of the large size of the individual farms, the total area accounts for half of the entire agricultural land in our case study region, the hotspot 29.
For more clarity, we present some of the characteristics of the investigated farms in table 1.
In table 2, we present the results of the interviews, outlining how many interviewees gave which answer/ referred to which factor, and also included the percentages. In the discussion, we outlined what the biggest problems were.
The measure with the legumes is an official measure under the current agri-environmental program – the species to be used are not specified.
Reviewer 2 Report
Comments and Suggestions for Authors
Dear authors,
I find your paper Turning the tide: an analysis of factors influencing the adoption of biodiversity-enhancing measures on agricultural land at the German Baltic coast to be a very nice piece of qualitative research.
The paper is well structured and the research is well designed.
I have a few comments and suggestions. I would appreciate it if you could take them into consideration.
I would be grateful for some statistical data on the environmental indicators at the study site (CAP Context Indicators).
What kind of policy measures are there in the case study? I am referring to the Common Agricultural Policy rural development measures, in particular the so-called IACS measures.
How much support was available in the last programming period (2014-2020) and what impact did it have? Could you describe in more detail how Potential Solutions could be integrated into the current/existing measures?
The methodology is based on the model of biodiversity-enhancing behavior according to Stoll-Kleemann (2019). Has your research contributed to improving the model and how? Please explain.
Author Response
We appreciate the careful review and would like to thank the reviewer for the comments. We feel that they have clarified some aspects of the manuscript.
The article includes CAP context indicators, namely land cover and the percentage of FFH areas. We have now also included farming intensity as another important aspect. Other CAP context environmental indicators are not available on state level (Mecklenburg-Vorpommern), and have therefore not been included.
We have included a description of the currently valid measures under pillar 1 and pillar 2 as far as they are relevant to biodiversity conservation.
We have included the amount of funding paid under the last period. An impact evaluation has not been made, and we are therefore unable to answer this question. We have outlined our hope that the potential solutions we have outlined will be taken up under the next funding period.
We have explained in the Methodology section that the model has not been improved, but adapted to reflect the specific conditions of our case study.
Reviewer 3 Report
Comments and Suggestions for Authors
This manuscript is well organized, and the drawn conclusions are coherent with the obtained results. The paper was well written!
Keywords: Please, arrange the keywords alphabetically.
Introduction: I think that you should these important references as examples to support your sentence: “Many landscapes were converted into intensively farmed uniform large-scale feedlots, which are a major threat to farmland wildlife such as birds and insects.”. I would like to suggest:
Fraissinet, M., et al., (2023). Responses of avian assemblages to spatiotemporal landscape dynamics in urban ecosystems. Landscape Ecology, 38(1), 293-305.
Arellano, L., et al., (2023). Dung beetles (Coleoptera: Scarabaeidae) in grazing lands of the Neotropics: A review of patterns and research trends of taxonomic and functional diversity, and functions. Frontiers in Ecology and Evolution, 11, 1084009.
Introduction: Please, explain in detail you hypothesis and predictions.
Introduction: Please, add the north symbol and the scale in the figure 1.
Materials and Methods: The paragraph personal factor should be reduced.
Results: Well written!
Discussion and potential solutions: this two sections should be merged.
Comments on the Quality of English LanguageThe paper was well written!
Author Response
We would like to thank the reviewer for the helpful comments, our individual responses are listed below.
Keywords have been rearranged alphabetically.
Both references have been included.
Our hypothesis and the predictions have been explained.
The north symbol has been included, and a scale was included.
The paragraph has been slightly reduced – a further reduction would impact on the understanding of important terms. We therefore feel that the current length is required.
Discussion and potential solutions have been merged.
Reviewer 4 Report
Comments and Suggestions for Authors
Thank you very much for the opportunity to review this manuscript, the manuscript is interesting and the discussion on agricultural biodiversity is very interesting, but there are some additions that need to be added to make the article more understandable and complete.
1. The manuscript should be more detailed about the survey samples, I know very little about the survey samples, especially their individual characteristics and farm informations.
2. The authors' sample is too small, and they are all concentrated in the same area near the Baltic Sea in Germany, which is insufficient for the analysis of social and group factors, especially the inability to compare political frameworks with economic factors.
3. Can the author provide the results of the biodiversity assessment? Have the biodiversity measures taken achieved the desired objectives? Will the achievement of these goals also have an impact on the adoption of biodiversity measures by farmers?
4. How does the biodiversity crisis message reach farmers, and how does this affect farmers' planting decisions?
In short, this manuscript should strive to return from an investigative report to academic research, otherwise its academic significance and enlightenment role will be very limited.
Author Response
We would like to thank the reviewer for the helpful comments, which we have addressed as far as possible. The details are given below:
1. We are aware of the sample size – however, as a result of the large farm size in the area, the results are relevant for 23% of the hot spot area and almost 50% of its agricultural land. With this qualitative approach we aimed at displaying hotspot 29-specific characteristics of the farmers. Using a qualitative approach, we did not intend to find statistically significant correlations – instead, we intended to gather in depth information on factors relevant for decision-making.
2. The samples are purposefully concentrated in that area near the Baltic Sea – this is our case study region. We were particularly interested in finding out what the situation in that area is, and how it could be improved. We believe that this is what the results do. We also did not intent to compare political frameworks with economic factors – we tread both as factors and potential barriers for implementing biodiversity-enhancing measures.
3. A biodiversity assessment has not been part of our research. Nevertheless, we tried to find biodiversity assessments for the investigated region for the last GAP-period and did not find adequate ones. In fact, no evaluation of the effectiveness on the biodiversity-enhancing measures has been made by anyone, and we have stated this in the paper. Within the frame of this qualitative research, we have identified factors that play a role for the uptake of biodiversity-enhancing measures, and based on this, suggested potential improvements/ solutions.
4. We have not specifically addressed this question in our research, but the interviews provided some indication on that aspect. We have included a respective statement as part of the results section (regarding the French wheat variety).
In short, this manuscript should strive to return from an investigative report to academic research, otherwise its academic significance and enlightenment role will be very limited.
We believe that with the additional information provided, the paper does offer some interesting insights that could help to improve the upcoming GAP funding period.
Round 2
Reviewer 1 Report
Comments and Suggestions for Authors
The manuscript is significantly improved and can be accepted for publication.
Author Response
We appreciate the comment of the reviewer and like to thank for the recommendation to publish the paper.
Reviewer 4 Report
Comments and Suggestions for Authors
Thank you very much to the author for the response to the comments I raised. In the reply, the author believes there is did not intent to compare political frameworks with economic factors. However, in the discussion section, an analysis is conducted on the content of the political framework and economic factors, such as bureaucracy and land tenure, all of which have a significant impact on the behavior of farmers under a changing policy environment.
Author Response
We thank the reviewer for the possibility to explain our argument. Yes, it is correct that we outline factors such as the political framework and economic factors and their influence of the behavior of farmers. However, we do not intent to provide a statistical significant correlation of these factors with the decision-making behavior, and our comment just refered to this fact. We hope we were able to clarify this point.